

# Revising short and longwave radiation archives in view of possible revisions of the WSG and WISG reference scales: Methods and implications

Stephan Nyeki[1], Stefan Wacker[2], Julian Gröbner[1], Wolfgang Finsterle[1], and Martin Wild[3]

[1]Physikalisch Meteorologisches Observatorium and World Radiation Centre (PMOD/WRC), Davos, Switzerland
[2]Asiaq, Greenland Survey, Nuuk, Greenland
[3]IACETHZ, Zürich, Switzerland

*Correspondence to*: Stephan Nyeki (stephan.nyeki@pmodwrc.ch)

**Abstract.** A large number of radiometers are traceable to the World Standard Group (WSG) for shortwave radiation and the interim World Infra-red Standard Group (WISG) for longwave radiation, hosted by the Physikalisch Meteorologisches Observatorium Davos/World Radiation Centre (PMOD/WRC, Davos, Switzerland). The WSG and WISG have recently been found to over- and underestimate radiation values, respectively (Fehlmann et al., 2012; Gröbner et al., 2014), although research is still ongoing. In view of a possible revision of the reference scales of both standard groups, this study discusses the methods involved, and the implications on existing archives of radiation time-series, such as the Baseline Surface Radiation Network (BSRN). Based on PMOD/WRC calibration archives and BSRN data archives, the downward longwave radiation (DLR) time-series over the 2006 – 2015 periods were analysed at four stations (polar and mid-latitude locations). DLR was found to increase by up to 3.5 and 5.4 W m$^{-2}$, respectively, after applying a WISG reference scale correction and a minor correction for the dependence of pyrgeometer sensitivity on atmospheric integrated water vapour content. Similar increases in DLR may be expected at other BSRN stations. Based on our analysis, a number of recommendations are made for future studies.

## 1 Introduction

In order to ensure the world-wide homogeneity and calibration of radiation measurements, the World Radiation Centre at the Physikalisches Meteorologisches Observatorium Davos (PMOD/WRC) in Davos (DAV; 46.82°N, 9.85°E, 1580 m asl; Switzerland), was established on behalf of the World Meteorological Organisation (WMO). The World Standard Group (WSG) of pyrheliometers was used to establish the World Radiometric Reference (WRR) in 1970, and represents the SI unit of radiation for the shortwave range. The latter is broadly defined as covering the wavelength range ~0.3 – 3 μm while the WSG pyrheliometers cover the ~0.25 – 4 μm range. The corresponding standard group for longwave radiation, the World Infrared Standard Group (WISG) of pyrgeometers, was established in 2004 on the recommendation of the WMO (WMO, 2003) but is an interim working standard due to a number of ongoing issues (Gröbner et al., 2014; Philipona, 2015). Longwave radiation is broadly defined as covering the ~4 – 100 μm range while the WISG pyrgeometers cover the range ~4 – 50 μm (Eppley PIR pyrgeometers) and ~4.5 – 42 μm (Kipp&Zonen CG(R)4 pyrgeometers). Recent measurements with newly developed high-precision ground-based radiometers have demonstrated that a revision of approximately -0.3% and up to +5 W m$^{-2}$ of the WSG and WISG scales, respectively, may be required (Fehlmann et al., 2012; Gröbner et al., 2014). Since a large number of short and longwave radiation time-series (e.g. from the Baseline Surface Radiation Network; BSRN; Ohmura et al., 1998) are traceable to the WSG and WISG, these may also need to be revised. If this is the case, then there will undoubtedly be a number of challenging issues as records are used to validate/calibrate satellite surface products and climate model outputs.



In order to address these issues, the Commission for Instruments and Methods of Observations (CIMO) proposed that Task Teams should be established: 1) to assess the consequences of a revision of the WSG and WISG reference scales with regard to BSRN, 2) to make recommendations for a modification of the current reference scales, and 3) to propose methods on how to deal with archived BSRN data (CIMO, 2013).

The objective of this study is to use data from the PMOD/WRC and BSRN archives (up to December 2015) to address these issues. The implications of a revision, based on the study by Gröbner et al. (2014), are then considered with respect to the BSRN archive which is followed by an initial assessment of the effect on downward direct shortwave and downward longwave radiation (DSR, DLR) time-series at three BSRN stations (Georg von Neumayer, GVN; Ny Ålesund, NYA; Payerne, PAY) and at Davos (DAV). Conclusions as well as recommendations for further studies are then presented

in the final section.

## 2 Methods and data

A brief overview of the WSG and WISG standard groups, and the BSRN archive will be given in this section. Possible methods to re-calibrate radiometers with respect to new WSG and WISG reference scales, and methods to revise the BSRN archive will also be discussed.

**2.1 The World Standard Group (WSG) for shortwave radiation: A brief overview**

The WSG currently consists of six pyrheliometers to measure direct broadband solar radiation (Fröhlich, 1991). While the long-term stability of the WRR is within an uncertainty range of ±0.3% (Finsterle, 2016), the absolute radiation of the WRR is 0.3% higher than the SI scale due to internal discrepancies in the WSG instruments which define the WRR (Fehlmann, 2012). Since the WSG was established in 1977, more than 300 pyrheliometers have been calibrated (e.g. 21 Eppley NIP, 26

K&Z CH1, 26 K&Z CHP1 etc) at PMOD/WRC. According to PMOD/WRC archives, virtually all pyrheliometers worldwide have a calibration traceable to the WSG due to several aspects: i) The broad acceptance of the WSG since its realisation, ii) five-yearly International Pyrheliometer Comparisons at the PMOD/WRC (Finsterle, 2016) and other regional and national comparisons, and iii) the use of a travelling standard by manufacturers from the onset.

The WSG accuracy was derived from a comprehensive comparison in which a radiometer traceable to the WRR

was compared to the primary laboratory radiometric standard at the National Physical Laboratory (NPL) in London, and the Total solar irradiance Radiometer Facility (TRF) at the Laboratory for Atmospheric and Space Physics (LASP) in Boulder (Fehlmann et al., 2012). The TRF is designed to directly compare a solar radiometer to a reference cryogenic radiometer calibrated at the National Institute of Standards and Technology (NIST).

The CIMO Task Team on Radiation References is currently considering recommendations on how to harmonise the

WRR with SI laboratory standards. A likely scenario will include decommissioning of the current WSG in favour of cryogenic solar radiometers, such as the Cryogenic Solar Absolute Radiometer (CSAR), to represent the WRR. The Task Team will publish its recommendations during the CIMO Session 17 to be held in 2018. For specific applications, i.e. to assess solar energy potential, network radiometers (pyrheliometers and pyranometers) which measure the direct beam of the sun, the total (global) shortwave radiation can be readily revised by lowering readings by 0.3% as proposed by Wild et al.

(2013) and based on the findings by Fehlmann et al. (2012). No additional parameters (e.g. atmospheric, instrumental, etc) are required for the revision.





### 2.2 World Infra-Red Standard Group (WISG) for longwave radiation

#### 2.2.1 A brief overview

The WISG consists of two Eppley (PIR-31463 and 31464) and two Kipp & Zonen (K&Z; CG4-010535 and FT004) commercial pyrgeometers which are operated continuously on the PMOD/WRC roof platform, and are individually referred

to as WISG-1 to 4, respectively. In addition, three other pyrgeometers have been simultaneously measuring alongside the WISG but are not officially constituents of it. These include: i) K&Z CG4-030669, since February 2004, ii) K&Z CGR4-110390 (dome without a solar-blind filter), since November 2011, and iii) Hukseflux IR20-105 since April 2012. The pyrgeometer CG4-030669 will be used later on in this study to represent this group due to its longer continuous time-series. Returning to WISG-1 and 4, these were originally calibrated with respect to the Absolute Sky Scanning Radiometer (ASR;

Philipona, 2001a) during the International Pyrgeometer and ASR comparison (IPASRC-I) in 1999 at the Atmospheric Radiation Measurement (ARM) Southern Great Plains site, Oklahoma, USA (Philipona et al., 2001b). The operational sensitivities of WISG-1 to 4 are 3.534, 3.585, 12.320, and 9.590 $\mu V\ W^{-1}\ m^2$, respectively (Gröbner et al., 2014). A WISG calibration with respect to IPASRC is referred to here as $WISG_{IPASRC}$. More than 230 pyrgeometers have been calibrated at PMOD/WRC since about 1992, initially with respect to a black-body source, and from 2004 onwards with respect to both

the WISG and black-body. When categorised by pyrgeometer type, the total of 230 consists of 122 Eppley PIR, 98 K&Z CG4/CGR4 and about 10 from other manufacturers.

The WISG has never been re-calibrated since IPASRC-I, and thus its traceability to SI units using the ASR has never been subsequently re-established and verified. While a pyrgeometer calibration with respect to the WISG is possible with a relative expanded uncertainty (95% coverage probability) of 0.9%, the WISG absolute uncertainty of ±2.6 W m⁻² is

limited by the traceability of the WISG to SI units. Regarding the internal stability of all four pyrgeometers comprising the WISG, this was demonstrated to be ±1 W m⁻² over the 2004 – 2013 period (Gröbner and Wacker, 2013)

Longwave radiation is calculated by PMOD/WRC using the so-called extended Albrecht & Cox equation (e.g. Philipona et al., 1995):

$$E = \frac{U}{C}(1 + k_1 \sigma T_B^3) + k_2 \sigma T_B^4 - k_3 \sigma (T_D^4 - T_B^4),\tag{1}$$

where $E$ is the longwave radiation in W m⁻², $U$ is the measured voltage of the pyrgeometer thermopile in volts, $C$ is the pyrgeometer sensitivity in V W⁻¹ m², $\sigma$ is the Stefan-Boltzmann constant (5.6704 x 10⁻⁸ W m⁻² K⁻⁴), $T_B$ and $T_D$ are the measured body and dome temperatures of the pyrgeometer in Kelvin, respectively, and $k_i$ are the instrument constants. In a standard pyrgeometer calibration procedure at PMOD/WRC, $k_i$ are determined in the laboratory using a reference black-body

while $C$ is retrieved relative to the WISG average from outdoor night-time measurements during clear-sky conditions (Gröbner and Wacker, 2015).

Customer pyrgeometers are routinely sent to be calibrated alongside the WISG while others are calibrated using a travel standard pyrgeometer which itself has been calibrated with respect to the WISG. These will be referred to as "direct" and "indirect" calibrations, respectively.

#### 2.2.2 Evidence for a revision of the WISG reference scale

The WISG is currently regarded as an interim transfer standard group with respect to its reference scale (WMO, 2006). The realisation of a more accurate standard group to determine irradiance (Reda et al., 2012) and a revision of the WISG scale have been ongoing issues in recent years (e.g. Gröbner et al., 2014; Philipona, 2015; Gröbner et al., 2015). However, a replacement of the WISG as a transfer standard is not foreseen due to its all-weather and hence continuous measurement



capabilities. Current state-of-the-art research radiometers such as the Infra-Red Integrating Sphere radiometer (IRIS; Gröbner, 2012) and the Absolute Cavity Pyrgeometer (ACP; Reda et al., 2012) are window-less and thus are not suitable for continuous all-weather operation.

The evidence for a revision of the reference scale comes from concurrent operation of the WISG alongside IRIS
during night-time clear-sky conditions since 2008 which yielded an underestimation of the WISG clear-sky longwave irradiance by $2 - 6$ W m$^{-2}$, depending on the amount of integrated water vapor (IWV) (Gröbner et al., 2014). These results have been confirmed in two inter-comparison campaigns with the ACP (Gröbner et al., 2014) where the ACP and IRIS measurements were consistent to within ±1 W m$^{-2}$ during both campaigns (which is within the instrumental uncertainties of ±4 W m$^{-2}$ and ±2 W m$^{-2}$, respectively,) while the WISG measured lower values by an average of 5.6 W m$^{-2}$ (Gröbner et al.,
2014). Further support comes from measurements during the second International Pyrgeometer Comparison in 2015 at PMOD/WRC (unpublished data). However, it should be mentioned that Philipona (2015) questioned the small number of simultaneous measurements, various technical issues, and highlighted various other important aspects with respect to the IPASRC campaigns. In addition, it was recommended that further large-scale inter-comparisons should take place before changing the WISG reference scale.

Based on simultaneous WISG and IRIS measurements since 2011, Gröbner et al. (2014) recommended that $C$ values of WISG-1 to 4 should increase by an overall average of 6.5% to 3.798, 3.791, 13.192, and 10.139 μV W$^{-1}$ m$^2$, respectively. A calibration of the WISG with respect to IRIS is referred to here as WISG$_{IRIS}$. The $k_i$ constants, in this case, are the same as before. However, a revision of the reference scale cannot be linearly applied to $E$ as: i) Eq. 1 is non-linear with respect to $C$, and ii) $C$ shows a dependence on IWV (section 2.2.3). The re-calibration of customer pyrgeometers and
radiation time-series is therefore somewhat more involved. A re-calibration of customer pyrgeometers which have previously been at PMOD/WRC would require archived calibration data (available from 2004 to present) of $U$, $T_B$ and $T_D$ to firstly determine a new $C$ value. In a second step, radiation time-series from customer pyrgeometers would then be re-calculated by using the new $C$ value and archived $U$, $T_B$ and $T_D$ from the station. If a pyrgeometer is not traceable either directly or indirectly to the WISG, then a re-calibration and hence re-calculation of the radiation time-series is not possible.

**2.2.3 Dependence of sensitivity $C$ on atmospheric integrated water vapour content**

When pyrgeometers are calibrated outdoors alongside the WISG at PMOD/WRC, enough valid calibration data are typically collected after periods up to 4 weeks, depending mainly on the weather. On occasion, users request a longer calibration period spanning three seasons, a so-called 3-season calibration, in order to achieve a more accurate sensitivity value. It was observed that the sensitivity $C$ of certain groups of pyrgeometers showed a dependence on atmospheric IWV depending on
whether a calibration with respect to the WISG or CG4-030669 was used (Gröbner and Wacker, 2013; Gröbner et al., 2014). This was attributed to the spectral characteristics of pyrgeometer domes, observed in a previous study (Gröbner and Los, 2007). More specifically, $C$ was found to decrease with decreasing IWV when ≲ 10 mm for Eppley PIR and pre-2003 K&Z CG4 groups of pyrgeometers (referred to here as groups 1 and 2) when referenced against the PMOD/WRC pyrgeometers CG4-030669. However, a similar dependence was not observed in the group containing post-2003 K&Z CG4/CGR4 as of
serial number 030646 (group 3), according to K&Z and PMOD/WRC archives. On the other hand, when referenced against the WISG or individual pyrgeometers of the WISG, the above behaviour was reversed i.e. Eppley PIR and pre-2003 K&Z CG4 showed no significant dependence on IWV, while post-2003 K&Z CG4/CGR4 did. Common to all three groups was that no significant dependence was observed when IWV > 10 mm, regardless of the reference used. The cause of this behaviour remains to be investigated in a thorough scientific manner but it is thought to be due to the spectral transmission
characteristics of pyrgeometer domes (Gröbner and Los, 2007; Gröbner and Wacker, 2013). However, the important question is which group of pyrgeometers measures correctly when IWV < 10 mm? Evidence from IRIS, ACP and WISG





inter-comparisons (Gröbner et al., 2014) at PMOD/WRC would seem to suggest that the post-2003 group of K&Z pyrgeometers exhibits no significant dependence on IWV.

In order to avoid the apparent complication of a dependence of $C$ on IWV for the present, measurement points during calibration at PMOD/WRC are only considered valid when atmospheric IWV > 10 mm, amongst other quality control

criteria (Gröbner and Wacker, 2015). This limits the calibration season at PMOD/WRC from about March to November, and has been implemented since April 2012.

Regardless of which group of pyrgeometers exhibits a dependence on IWV, we will discuss the methods involved here and later on, the implications regarding longwave radiation time-series in section 3. In order to correct longwave radiation time-series, concurrent IWV data at the ground station from radiosonde ascents, microwave radiometry or GPS, is

required. For those BSRN stations without these measurements, IWV can be derived from re-analysis data (e.g. ERA-40) or one of many empirical models using meteorological data (temperature and relative humidity at 2 m above ground, $T_{2m}$ and $RH_{2m}$). IWV either from GPS or an empirical model (Leckner, 1978) was favoured in this preliminary study mainly due to: i) the ready availability of data, ii) the high data resolution (1 – 10 min.), and iii) the length of the GPS and meteorological time-series. GPS IWV was used for DAV and PAY, and modelled IWV for GVN and NYA. The uncertainty in IWV from

these various methods is estimated at ~1 mm which corresponds to an uncertainty in $C$ of ~0.6% when IWV < 10 mm. Using the criterion that a 1% uncertainty in $C$ is acceptable, then a maximum uncertainty of ~1.7 mm in IWV would still be acceptable.

The characterisation of $C$ as a function of IWV can be determined during a three-season outdoor calibration at PMOD/WRC which usually takes about 6 months. Sending a pyrgeometer to PMOD/WRC for this length of time is

logistically difficult for most stations, so a more practical approach would be the use of a general empirical correction which could be generally applied. This will be discussed later in section 3.

**2.3 The BSRN archive**

BSRN (Ohmura et al., 1998; McArthur, 2005) holds the world's most accurate archives of radiation data which are used to validate satellite products and the radiation budget of the Earth-atmosphere system (Trenberth et al., 2009; Stephens et al.,

2012; Wild, 2012; Wild et al., 2013). All major climate zones are represented by 50+ stations which are currently in the BSRN network. Radiation measurements, collocated surface and upper-air meteorological observations, and station metadata are archived in an integrated database (http://bsrn.awi.de). Despite continuous efforts, uncertainties in the determination of individual components of the surface radiation budget still exist (Wild, 2017). An improvement in the accuracy of BSRN time-series may therefore help to reduce such uncertainties.

**2.4 BSRN time-series**

BSRN downward short and longwave time-series were revised for a small selection of stations, including NYA, GVN, PAY and DAV (see Table 1 for site details). The first three stations belong to the BSRN network amongst others, and were mainly chosen due to: i) direct traceability of all pyrgeometers to the WISG with regular calibration every 2 – 4 years, ii) the ready availability of pyrgeometer raw data (i.e. $U$, $T_B$ and $T_D$), and iii) the length (10+ years) and continuity of the DLR time-

series. Although a number of other BSRN stations fulfil the above points, none were able to readily provide raw data, including stations at low-latitudes. This highlighted a number of important issues regarding the availability of obtaining current or historical raw data. For instance: i) stations may often have too few personnel resources, ii) a knowledge-pool may no longer exist due to retirement, and iii) data may not be readily accessible due to software/hardware legacy issues. In our case, we were able to obtain raw pyrgeometer data (1-min resolution) for PAY and DAV while DLR and pyrgeometer





temperatures were obtained for GVN and NYA. Through use of Eq. 1, it was possible to determine the original raw pyrgeometer voltage (1-min resolution) for the latter two stations.

BSRN time-series correspond to so-called "all-sky" conditions (i.e. clear and cloudy conditions) but for climatological studies, clear-sky conditions are also of importance. For instance, a revised DLR reference scale will have a

5 proportionally larger effect on DLR time-series at stations with a higher clear-sky fraction. In order to determine DLR time-series during clear-sky conditions, 2-m meteorological data (10-min resolution) from each station was used to calculate the partial cloud amount (PCA) with the APCADA algorithm (Dürr and Philipona, 2004). The algorithm uses $T_{2m}$, $RH_{2m}$ and DLR together with a set of empirical rules to calculate the PCA at any time of day. PCA was calculated with a 1-min resolution, and values ≤1 corresponded to clear-sky conditions. While the PCA can be reliably determined for low and mid-

10 altitude clouds, APCADA is less efficient for high-altitude clouds. However, for the purposes of our comparative study, APCADA is considered to be satisfactory (e.g. Gröbner and Wacker, 2011).

Before revised DLR time-series were calculated, extensive tests were conducted to ensure that the existing PMOD/WRC calibration software (Gröbner and Wacker, 2015) was able to re-calculate previous $C$ values, and that BSRN DLR time-series could be precisely reproduced for all four stations in this study.

## 3 Results and discussion

This section discusses the results from an analysis of short and longwave radiometers in the PMOD/WRC and BSRN archives. The focus is mainly on the latter as a possible revision of the WSG reference scale will not be as complex a task as the WISG scale.

### 3.1 PMOD/WRC archives: Calibration frequency of pyrgeometers

As mentioned previously, over 230 pyrgeometer calibration records are in the PMOD/WRC archives. To date, 58 are being used by BSRN and 73 by other users. Unfortunately, the calibration history of many pyrgeometers is difficult to assess although details have been recorded in the PMOD/WRC archive whenever available. Records indicate that of the 73 pyrgeometers (PIR and CG4/CGR4) not being used by BSRN, at least 39 (14, 11, 8, and 1) have been calibrated once (twice, three, four, and five times) against the WISG. The average period between calibrations was 4.0 years with a minimum of

about 1 and maximum of 10 years. BSRN pyrgeometers are considered further below.

### 3.2 PMOD/WRC archives: Pyrgeometer sensitivity $C$ as a function of IWV

While the dependence of $C$ on IWV has been previously reported for several pyrgeometers (Gröbner and Wacker, 2013), a detailed analysis of previous pyrgeometer calibrations in the PMOD/WRC archive during the present study was carried out to obtain a better overview. It was found that a total of 27 pyrgeometers (including WISG-1 to 4) have measured

continuously for at least 90 days but only 14 have measured over the 2 – 25 mm IWV range. These are listed in Table 2 along with a further three pyrgeometers which have sufficient measurements over the 2 – 15 mm IWV range but during non-continuous periods. Pyrgeometers are divided into the three groups defined in section 2.2.3, namely: Eppley PIR, pre-2003 K&Z CG4, and post-2003 K&Z CG4/CGR4. Records indicate that only one of the listed pyrgeometers has participated in BSRN, while most of the others are travelling standards for meteorological/governmental institutes. Relatively few

measurements for Eppley pyrgeometers are available when considering the large number in worldwide use, especially within BSRN. This is probably due to the fact that most Eppley PIR have been calibrated at regional centres in North America using pyrgeometer standards traceable to the WISG.



Graphs of $C$ as a function of IWV from each of the three pyrgeometer groups are shown in Figure 1a-f, and are calibrated with respect to the WISG and CG4-030669 in the left and right columns, respectively. All graphs show that $C$ is essentially a constant when IWV ≥ 10 mm but below this value, $C$ exhibits a distinct decrease in Figure 1b and 1d, and an increase in Figure 1e. This behaviour has previously been characterised with a linear model (Gröbner et al., 2014; Gröbner

and Wacker, 2015). Various other models were tested here, including a $5^{th}$ order polynomial fit, however, similar results to the linear model were obtained when applied to irradiance time-series. The choice of IWV = 10 mm as the inflection point was based in the above previous studies on an empirical analysis. When examined using differential analysis, the inflection point was found to vary in the IWV ~ 8 – 12 mm range, with an average value at about 10 mm. For the sake of consistency, IWV = 10 mm is therefore also used here.

Table 2 lists the slope values of $C$ with respect to the WISG and CG4-030669 when IWV < 10 mm for individual as well each group of pyrgeometers. Average slope values in each group are shown in bold, and exhibit either: i) elevated values (0.024, 0.066 and -0.054 $\mu V\ W^{-1}\ m^2$ per mm IWV for groups 1, 2 and 3, respectively) representing a distinct dependence of $C$ on IWV or ii) values close to zero (-0.001, 0.004 and 0.008 for groups 1, 2 and 3) when virtually no dependence of $C$ on IWV occurs. In order to assess these results better, relative slope values in percent are also shown in

Table 2. For the Eppley group, $C$ is ~4.9% (0.61 x 8) lower with respect to CG4-030669 when IWV = 2 mm. Similarly, $C$ is also ~4.9% lower for the pre-2003 K&Z group while the post-2003 K&Z group is ~3.9% higher with respect to the WISG. Values of IWV as low as 1 – 2 mm are generally representative of polar (e.g. GVN and NYA) and high-alpine stations (DAV), so a relative increases/decrease in $C$ of up to 5% can be regarded as a maximum value.

These results support earlier observations and conclusions (Gröbner and Los, 2007; Gröbner and Wacker, 2013,

Gröbner et al., 2014) that certain types of pyrgeometer domes and/or their coatings may be responsible for the observed dependence of $C$ on IWV. A general slope value cannot yet be defined with any confidence for the pre-2003 K&Z CG4 group as only two have been characterised in Table 2 but values for the Eppley PIR and post-2003 CG4/CGR4 groups are considered to be provisionally representative. Nevertheless, there is still a clear necessity to characterise more pyrgeometers for up to 12 month periods. For instance, $C$ values of four pyrgeometers (one Eppley and three K&Z) could not be assigned

to any particular group i.e. there is simultaneously a weak dependence on IWV with respect to both a WISG and a CG4-030669 calibration. An explanation has not yet been found.

Now that general slope values of $C$ are available for each pyrgeometer group, they will be used in the next section to revise DLR time-series. Table 1 shows the pyrgeometers used at the respective stations. As none of these Eppley pyrgeometers have been individually characterised for long enough periods at PMOD/WRC, a group average slope value of

0.24 $\mu V\ W^{-1}\ m^2$ per mm IWV from Table 2 will be used. A single K&Z pyrgeometer (CGR4-110355) has been used to date to construct BSRN time-series at the three BSRN stations in Table 1. It is also the only BSRN field pyrgeometer that has had its $C$ versus IWV dependence fully characterised at PMOD/WRC. A slope value of 0.007 $\mu V\ W^{-1}\ m^2$ per mm IWV from Table 2 effectively means there is no dependence of $C$ on IWV.

### 3.3 BSRN archives: Traceability of short and longwave radiometers to the WSG and WISG

BSRN measurements aim to achieve the highest standards regarding accuracy, observational procedure, and calibration methods. As already mentioned, virtually all pyrheliometers are believed to be traceable to the WSG. However, the situation regarding pyrgeometers is not so clear. Only 58 BSRN pyrgeometers had been calibrated at PMOD/WRC up to December 2015 which have been used for monitoring at 15 BSRN stations (including PAY, GVN, and NYA amongst others). The calibration history of individual pyrgeometers is not well-documented in many cases but generally better than for non-BSRN

pyrgeometers. Of these 58, at least 21 (12, 9, 6, 3, 0, 1, 2, 1, and 3) have been calibrated once, (twice, three times … ten times) against the WISG. The average period between calibrations was 3.1 years with a minimum of about 1 and a maximum





of 18 years. Only a handful of these pyrgeometers therefore appear to have had a "regular" calibration according to our records. Whether this reflects the calibration history of pyrgeometers at institutes which use travelling standards, is at present unknown to us.

An overview of BSRN pyrgeometers and those with WISG traceability is shown in Table 3. Of a total of 223, 188
are Eppley PIRs and 35 are K&Z CG4/CGR4s. Although only 58 are directly traceable to the WISG, the number with indirect traceability to the WISG is estimated at 64. This latter number is a lower estimate but is probably representative of the current situation as all major meteorological or governmental institutes responded to a PMOD/WRC questionnaire, sent to gather data for the present study.

If pyrgeometers with a direct and indirect traceability are added together then at least 46% of Eppley PIRs and
100% of K&Z CG4/CGR4s are traceable. A maximum of 56% of Eppley PIRs may therefore have a different traceability, either to the original black-body calibration or to another calibrating institute. Further efforts would be useful to determine the traceability of these pyrgeometers as not all questionnaires sent to BSRN station personnel were returned. These findings therefore imply that a number of BSRN longwave radiation time-series may still be partially or fully based on calibrations not traceable to the WISG.

**3.4 BSRN archives: Application of possible new WSG and WISG reference scales to BSRN time-series**

Calibration histories of pyrgeometers at all four stations were used to re-calculate $C$ values with respect to $WISG_{IRIS}$ (see Table 1) and to calculate $C$ as a function of IWV according to the methods described in section 2.2.3. DLR time-series were then calculated for four scenarios: i) $WISG_{IPASRC}$ and $C \neq f(IWV)$ i.e. this corresponds to the current method used to calculate BSRN time-series, ii) $WISG_{IRIS}$ and $C \neq f(IWV)$, iii) $WISG_{IPASRC}$ and $C = f(IWV)$, and iv) $WISG_{IRIS}$ and $C =$
$f(IWV)$. Figure 2 illustrates DLR from 2006 – 2015 at NYA for scenario 1. Monthly mean DLR values for all-sky and clear-sky conditions are shown along with 1-min values for comparison. The seasonal DLR cycle exhibits a maximum in summer and minimum in winter as a result of seasonal temperature and humidity conditions. Although time-series extend back to the 1990s at all four stations, the 2006 – 2015 period was chosen to aid in comparing results as outdoor WISG calibrations of NYA and GVN pyrgeometers only began in 2006. The mean DLR for all-sky conditions at
NYA in Table 4 shows that scenario 1 gives 258.7 W m$^{-2}$ while use of the IRIS scale in scenario 2 increases the mean by 2.4 W m$^{-2}$ to 261.1 W m$^{-2}$. A similar increase of 2.0 W m$^{-2}$ occurs at GVN while PAY is lower at 1.4 W m$^{-2}$ and DAV is higher at 4.2 W m$^{-2}$. The range of values can be explained by the relative frequency of clear-sky conditions at each location, defined here by the percentage of clear-sky to all-sky conditions. NYA and GVN experience clear-sky conditions 15% and 24% of the time, and PAY and DAV 5% and 29%, respectively. A higher percentage of clear-sky conditions results in a greater
fraction of the time-series being affected by a change of the WISG reference scale, and vice-versa.

Scenarios 3 and 4 are similar to 1 and 2, respectively, except that the sensitivity $C$ has been corrected to take the dependence on IWV into account. Figure 3 shows the NYA time-series of $C$ for scenario 4, for instance. While short periods of constant $C$ are visible, most other periods when IWV < 10 mm result in a variable value of $C$. When comparing scenario 3 to 1, and 2 to 4 in Table 4, this results in a reduction of DLR by 0.7 – 1.5 W m$^{-2}$ at all stations. However, of greater interest is
the overall effect of applying the WISG reference scale and IWV corrections (scenario 4) with respect to the current situation (scenario 1). Table 4 indicates that the increase is 1.3 and 0.7 W m$^{-2}$ for NYA and GVN, and 1.2 and 3.5 W m$^{-2}$ for PAY and DAV, respectively.

Although the correction of direct DSR time-series is straightforward, Table 3 shows the 2006 – 2015 average values at each station for the sake of completeness. All-sky direct DSR values in the final two columns are calculated: i) using the
BSRN archive, and ii) with a revision of -0.3%. Values range from 95.3 – 140.0 W m$^{-2}$ after application of the revision.

A similar analysis to Table 4 for clear-sky as opposed to all-sky conditions appears in Table 5. DLR at all stations in scenario 1 is lower by up to ~60 W m$^{-2}$ in comparison to values in Table 4, and illustrates the important effect that clouds





have on hindering the escape of longwave radiation to space. Again, scenario 4 is of greatest interest, and shows an increase in DLR of 2.2 and 1.8 W m$^{-2}$ for NYA and GVN, and 4.5 and 5.4 W m$^{-2}$ for PAY and DAV with respect to scenario 1. While these values are larger than those in Table 4, it should be mentioned that they only apply to clear-sky conditions which was shown earlier to occur < 29% of the time at all 4 stations. Regarding direct DSR clear-sky values, these are not shown in

Table 5 as there would just be a straightforward 0.3% reduction in values, similar to Table 4.

      If the analysis of DLR time-series in Tables 4 and 5 are considered as being representative of mid to high-latitude stations, then it implies that similar increases in average DLR may be expected. Unfortunately a low-latitude station could not be included in this study but a higher percentage of clear-sky conditions and higher IWV values, in particular, are likely to occur on average at such locations. The probable effect for all-sky and clear-sky conditions and scenario 4 would then be

an even larger increase in average DLR time-series than observed for PAY and DAV.

      A long-standing issue in climate models is a general underestimation of DLR when compared to BSRN-type surface observations (Wild et al. 1998, 2001, 2013). Although these low biases have generally decreased over time, increases in observed time-series as suggested above may imply that the underestimation of DLR continues to be a serious issue even in the latest generation of climate models used in the 5[th] IPCC assessment report (Wild et al. 2015). In the context of the

quantification of the global energy balance, estimates making use of the information contained in the surface observations (Ohmura and Gilgen, 1993; Wild et al. 1998, 2015) over many years have suggested a higher global mean DLR than typically advocated in various published global energy balance estimates such as those given in the IPCC assessments up to the 4[th] assessment report. An increase in observed DLR time-series may further support such higher DLR estimates within the global energy balance.

**4 Conclusions**

In view of a possible revision of the WSG and WISG reference scales, this study has discussed the methods involved, and the implications for existing BSRN archives of radiation time-series. However, updating archived data whether from individual stations or BSRN is not an easy task for a number of reasons. These aspects span a wide range of considerations from the availability of historical raw data, to the scientific benefits and then to the dissemination of revised data amongst

the wider user-community. While it is recognised that some of the logistical aspects involved are not trivial they would help to reduce the uncertainty in shortwave and longwave radiation BSRN time-series. Our conclusions can be summarised as follows:

1) Although not the focus of this study, the observed offset of the WISG and IRIS/ACP reference scales should be further

investigated by more independent and comprehensive inter-comparison measurements as previously suggested (Reda et al., 2012; Gröbner et al., 2014; Philipona, 2015). In this regard, several IRIS radiometers will be characterised in the immediate future and their traceability to SI units determined within the European Metrology Programme for Innovation and Research (EMPIR), together with partners from the metrology community. The aim of this project is to reduce the DLR uncertainty of IRIS radiometers to ±2 W m$^{-2}$.

2) PMOD/WRC and a questionnaire sent to BSRN station personnel indicate that a minimum of 46% of BSRN Eppley PIRs are either directly or indirectly traceable to the WISG while all BSRN K&Z CG4/CGR4s are traceable. Further coordinated efforts by manufacturers, calibration institutes, station personnel and end-users would be required to determine the traceability of the remaining 54% which could not be ascertained.





3) The dependence of the sensitivity $C$ on atmospheric IWV was investigated in greater detail in this study. Three groups of pyrgeometers were defined (Eppley PIR, pre-2003 K&Z CG4, and post-2003 K&Z CG4/CGR4) for which their dependence was empirically characterised. General empirical corrections for each group were determined but it is recommended that further extended comparisons should be conducted at PMOD/WRC and other sites in order to improve their accuracy.

4) The effect of revising the WISG reference scale increased the average all-sky DLR for the 2006 – 2015 period by 1.4 – 4.2 W m$^{-2}$ at the four stations (polar and mid-latitude locations) in this study. The increase was less (0.7 – 3.5 W m$^{-2}$) when the dependence of the sensitivity $C$ on IWV was also corrected. Average clear-sky DLR values were higher at 7.0 and 5.4 W m$^{-2}$. When considering other polar and mid-latitude BSRN stations, it can be reasonably argued that similar increases in DLR can be expected. Now that the methods in revising longwave radiation time-series have been defined and established, a more comprehensive future study focussing on the remaining 50+ BSRN stations would allow a more accurate assessment of the implications with regard to the global energy balance.

5) Regarding the submission of current data to BSRN, it is recommended that BSRN stations should not only continue to submit longwave radiation data but also raw pyrgeometer data (i.e. pyrgeometer signal voltage and temperature(s)) in the future. This would greatly simplify any possible revisions of longwave radiation time-series. Formal procedures and facilities to store this extra data in the BSRN archive were made several years ago but have not yet been used to our knowledge.

6) If historical time-series are to be revised then a re-submission to BSRN will also present its own difficulties. In the case of shortwave BSRN time-series, a scale revision applied by end-users may be simpler than the revision and re-submission of time-series by station personnel. On the other hand, a revision of longwave BSRN time-series is more difficult as it can only be applied to those pyrgeometers which are traceable to the WISG, and for which raw data are available. A future dedicated study within the framework of BSRN may be more effective in this case.

7) The potential re-definition of the WRR using a cryogenic radiometer (such as CSAR) is expected to imply a relatively trivial scale factor which transfers shortwave measurements form the old WRR regime to the cryogenic scale. Great care has to be taken in order to clearly attribute the applied (old or new) WRR scale to all existing and future shortwave measurements and archives.

*Acknowledgements.* The research work leading to this article was carried out as part of the Swiss National Science Foundation (SNF) project nos. 200021_157150. We thank Laurent Vuilleumier (MeteoSwiss, Switzerland), Marion Maturilli and Gert König-Langlo (AWI, Germany) for providing us with BSRN raw time-series data.

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




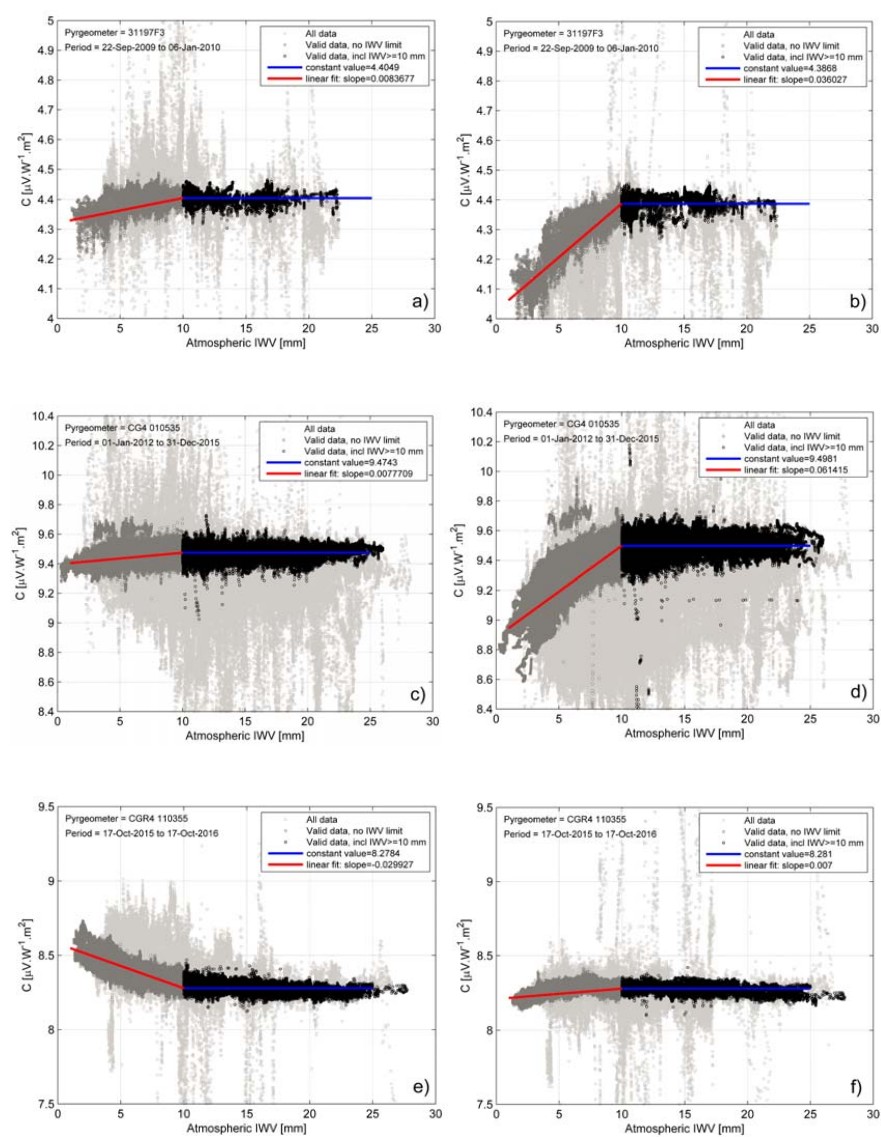

**Figure 1.** Graphs of sensitivity $C$ as a function of IWV during extended calibration measurements with respect to WISG$_{IPASRC}$ at PMOD/WRC. Rows contain graphs representative of the three groups discussed in the text: a-b) Eppley PIR group (PIR-31197), c-d) pre-2003 K&Z CG4 group (CG4-010535, i.e. WISG-4), e-f) post-2003 K&Z CG4/CGR4 group (CGR4-110355). Pyrgeometers in the left column are calibrated with respect to the WISG and those on the right with respect to CG4-030669.



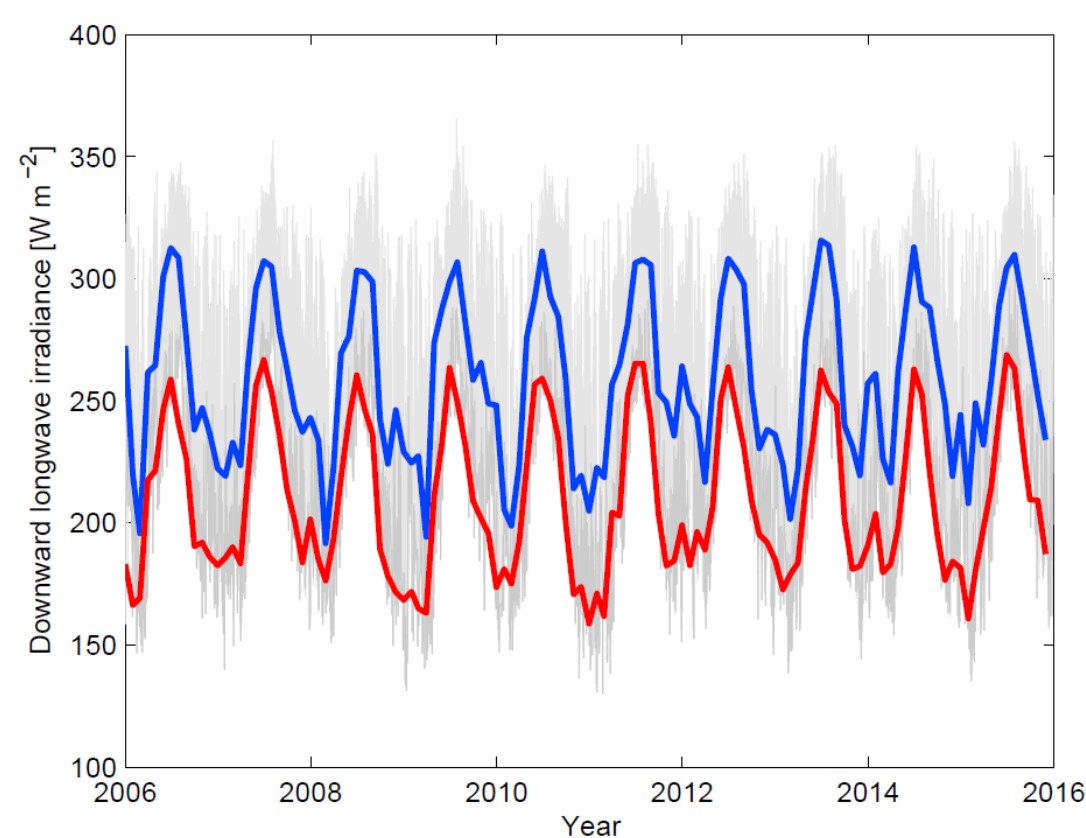

**Figure 2.** Monthly mean values of downward longwave radiation for all-sky (blue) and clear-sky (red) conditions for the 2006 – 2015 period at NYA. Corresponding all-sky (light grey lines) and clear-sky (dark grey lines) for 1-min values are shown in the background for comparison.





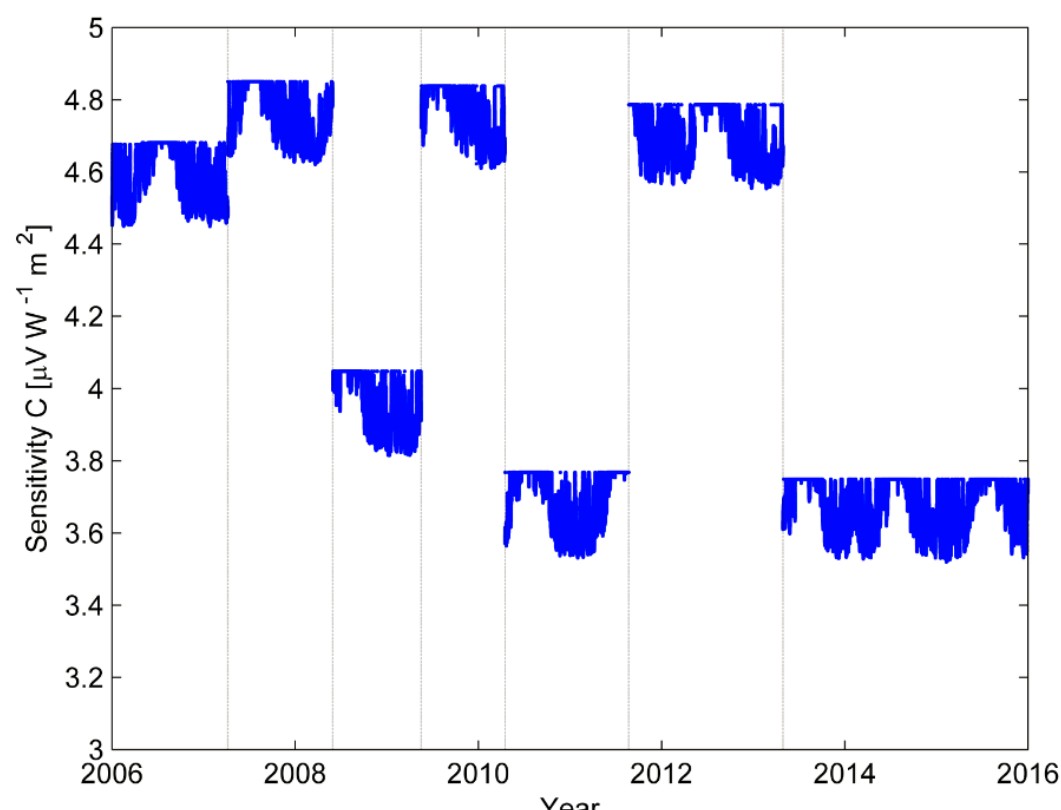

**Figure 3.** The time-series of pyrgeometer sensitivities $C$ used for scenario 4 to revise the 2006 – 2015 NYA DLR time-series. Vertical lines represent deployment periods of different pyrgeometers at NYA for which details are listed in Table 1.





10   **Table 1.** Details of the station locations, pyrgeometers used, and sensitivity $C$ values with respect to the WISG$_{IPASRC}$ and WISG$_{IRIS}$ reference scales.

| BSRN station | Period | Pyrgeometer type | $C$ [μV W$^{-1}$ m$^2$] with respect to WISG$_{IPASRC}$ | $C^b$ [μV W$^{-1}$ m$^2$] with respect to WISG$_{IRIS}$ |
|---|---|---|---|---|
| Ny Ålesund (NYA) | 1 Jan. 2006 – 2 May 2006 | PIR-28897 | 4.40 | 4.68 |
| 78.93°N, 11.93°E, | 2 May 2006 – 8 Apr. 2007 | PIR-28895 | 4.40 | 4.68 |
| 11 m asl, Svalbard | 8 Apr. 2007 – 30 May 2008 | PIR-28897 | 4.52 | 4.85 |
| | 30 May 2008 – 18 May 2009 | PIR-28895 | 3.82 | 4.05 |
| | 18 May 2009 – 17 Apr. 2010 | PIR-28897 | 4.55 | 4.84 |
| | 17 Apr. 2010 – 23 Aug. 2011 | PIR-28895 | 3.59 | 3.77 |
| | 23 Aug. 2011 – 29 Apr. 2013 | PIR-28897 | 4.55 | 4.79 |
| | 29 Apr. 2013 – 31 Dec. 2015 | PIR-28895 | 3.58 | 3.75 |
| Georg von Neumayer | 1 Jan. 2006 – 23 Jan. 2007 | PIR-28152 | 4.42 | 4.65 |
| (GVN), 70.65°S, 8.25°W | 23 Jan. 2007 – 14 Jan. 2008 | PIR-28150 | 4.57 | 4.83 |
| 42 m asl, Antarctica) | 14 Jan. 2008 – 17 Feb. 2009 | PIR-27603 | 3.76 | 3.98 |
| | 17 Feb. 2009 – 25 Jan. 2010 | PIR-29328 | 3.93 | 4.18 |
| | 25 Jan. 2010 – 26 Jan. 2014 | PIR-27600 | 3.39 | 3.59 |
| | 26 Jan. 2014 – 31 Dec. 2015 | PIR-28150 | 4.01 | 4.31 |
| Payerne (PAY) | 1 Jan. 2006 – 31 Mar. 2010 | PIR-28807 | 3.91 | 4.15 |
| 46.82°N, 6.94°E | 1 Apr. 2010 – 24 Mar. 2011 | PIR-31962 | 3.88 | 4.12 |
| 491 m asl, Switzerland) | 25 Mar. 2011 – 30 Sep. 2011 | PIR-29587 | 4.94 | 5.24 |
| | 1 Oct. 2011 – 31 Dec. 2015 | CGR4-110355 | 8.27 | 8.80 |
| Davos[a] (DAV) | 1 Jan. 2006 – 31 Dec. 2015 | PIR-31463 | 3.53 | 3.80[c] |
| 46.82°N, 9.85°E | | | | |
| 1580 m asl, Switzerland) | | | | |

[a]Note that DAV is not a BSRN station.

[b]The same $k_i$ constants for each period were used to re-calculate $C$.

15   [c]Value previously reported by Gröbner et al. (2014).



10 **Table 2.** Summary of pyrgeometers previously calibrated at PMOD/WRC with sufficient measurements to allow the dependence of $C$ on IWV to be characterised. Slope values of $C$ per mm IWV when IWV < 10 mm are shown with respect to the WISG and PMOD/WRC pyrgeometer CG4-030669. These results are also shown as relative values in percent. Numbers in brackets represent the standard deviation, while average values in bold are discussed in the text.

| Pyrgeometer group | Pyrgeo. S/N for period > 90 continuous days, over 2 – 25 mm IWV range | Pyrgeo.S/N for short non-continuous periods, over 2 – 15 mm IWV range | Slope of $C$ [μV W$^{-1}$ m$^2$] per mm IWV when IWV < 10 mm, with respect to: | | Relative slope of $C$ [%] per mm IWV when IWV < 10 mm, with respect to: | |
|---|---|---|---|---|---|---|
| | | | WISG | CG4-030669 | WISG | CG4-030669 |
| Eppley PIR | 29434 | | >0.000 | 0.028 | >0.00 | 0.70 |
| | 31197[a] | | 0.008 | 0.036 | 0.19 | 0.82 |
| | 31463[b] | | 0.005 | 0.025[l] | 0.14 | 0.70 |
| | 31464[c] | | -0.007 | 0.014[l] | -0.20 | 0.40 |
| | | 29255[k] | -0.002 | 0.024 | -0.03 | 0.51 |
| | | 29257[k] | -0.007 | 0.019 | -0.19 | 0.49 |
| | | 29258[k] | -0.002 | 0.022 | -0.06 | 0.65 |
| | **Avg. = -0.001 (0.006)** | | **Avg. = 0.024 (0.007)** | | **Avg. = -0.02** | **Avg. = 0.61** |
| Pre-2003 K&Z CG4 | FT004[d] | | 0.001 | 0.71[l] | 0.00 | 0.58 |
| | 010535[e] | | 0.008 | 0.061[l] | 0.08 | 0.64 |
| | **Avg. = 0.004 (0.005)** | | **Avg. = 0.066 (0.007)** | | **Avg. = 0.04** | **Avg. = 0.61** |
| Post-2003 K&Z CG4 and CGR4 | 010536[f] | | - 0.047 | -0.001 | -0.53 | -0.02 |
| | 030669[g] | | -0.071 | - | -0.58 | - |
| | 070037[h] | | -0.062 | 0.009 | -0.42 | 0.06 |
| | 070038[h] | | -0.067 | 0.004 | -0.57 | 0.03 |
| | 070039[h] | | -0.077 | 0.015 | -0.65 | 0.13 |
| | 100280[i] | | -0.049 | 0.005 | -0.43 | 0.04 |
| | 110355[j] | | -0.030 | 0.007 | -0.36 | 0.08 |
| | 110390[g] | | -0.030 | 0.015 | -0.34 | 0.17 |
| | **Avg. = -0.054 (0.018)** | | **Avg. = 0.008 (0.006)** | | **Avg. = -0.49** | **Avg. = 0.07** |

[a] NREL travelling standard, [b] WISG-1, [c] WISG-2, [d] WISG-3, [e] WISG-4, [f] K&Z travelling standard, new dome since 2005, [g, j] Since 2008 and 2011 alongside WISG, respectively, [h] JMA travelling standard, [i] K&Z travelling standard, [j] MeteoSwiss Payerne, BSRN station, [k] SURFRAD travelling standard, [l] Updated slope values for the period to Dec. 2015 reported here are similar to those previously by Gröbner et al. (2014) except for FT004 and CG4-010535 which were 0.100 and 0.075, respectively.



**Table 3.** The number of pyrgeometers which have submitted longwave irradiance time-series in the past to the BSRN archive are shown, as well as the number which have a direct or indirect traceability to the WISG.

|  | Eppley PIR | K&Z CG4/CGR4 | Total |
|---|---|---|---|
| N | 188 | 35 | 223 |
| N (direct traceability) | 47 | 11 | 58 |
| N (indirect traceability) | 40 | 24 | 64 |
| N (all traceability) | 87 | 35 | 122 |
| % (all traceability) | 46 | 100 | 55 |





**Table 4.** Mean and median (in brackets) values of the 2006 – 2015 *all-sky* DLR and direct DSR time-series for three BSRN (NYA, GVN, and PAY) stations using time-series from the BSRN archive, and for Davos (not a BSRN station). Four scenarios are shown where: i) the term "WISG$_{IPASRC}$" or "WISG$_{IRIS}$" refers to a WISG calibration based on the IPASRC campaigns or IRIS pyrgeometers, respectively, and ii) the pyrgeometer sensitivity $C$ is either a function or not a function of IWV. Scenario 1 represents the current BSRN archived time-series using current PMOD/WRC sensitivity values. Bold numbers represent the difference in mean and median (in brackets) DLR values with respect to scenario 1. Direct DSR median values are close to zero as a result of night-time measurements being included in the time-series calculation.

| BSRN station | DLR, mean (median) [W m$^{-2}$] | | | | Direct DSR, mean [W m$^{-2}$] | |
|---|---|---|---|---|---|---|
| | Scenario 1 WISG$_{IPASRC}$ $C \neq$ f(IWV) | Scenario 2 WISG$_{IRIS}$ $C \neq$ f(IWV) | Scenario 3 WISG$_{IPASRC}$ $C =$ f(IWV) | Scenario 4 WISG$_{IRIS}$ $C =$ f(IWV) | WSG$_{Current}$ | WSG$_{Revised}$ |
| Ny Ålesund (NYA) | 258.7 (266.1) | 261.1 (268.6) **2.4 (2.5)** | 257.5 (265.7) **-1.2 (-0.4)** | 260.0 (268.2) **1.3 (2.1)** | 159.2 (1.5) | 154.4 (1.5) |
| Georg von Neumayer (GVN) | 216.6 (220.6) | 218.6 (222.2) **2.0 (2.2)** | 215.1 (219.8) **-1.5 (-0.8)** | 217.3 (221.5) **0.7 (0.9)** | 162.3 (0.6) | 157.4(0.6) |
| Payerne (PAY) | 314.1 (317.4) | 315.5 (318.4) **1.4 (1.0)** | 313.9 (317.5) **-0.2 (-0.1)** | 315.3 (318.5) **1.2 (1.1)** | 83.1 (0.0) | 80.6 (0.0) |
| Davos[a] (DAV) | 280.9 (285.8) | 285.1 (289.5) **4.2 (3.7)** | 280.1 (285.6) **-0.8 (-0.2)** | 284.4 (289.3) **3.5 (3.6)** | 155.7 (1.0) | 151.0 (1.0) |

20    [a]Note that DAV is not a BSRN station.



**Table 5.** Same as Table 4 except for the 2006 – 2015 *clear-sky* DLR time-series.

| BSRN station | DLR, mean (median) [W m$^{-2}$] | | | |
|---|---|---|---|---|
| | Scenario 1 WISG$_{IPASRC}$ $C \neq$ f(IWV) | Scenario 2 WISG$_{IRIS}$ $C \neq$ f(IWV) | Scenario 3 WISG$_{IPASRC}$ $C =$ f(IWV) | Scenario 4 WISG$_{IRIS}$ $C =$ f(IWV) |
| Ny Ålesund | 201.8 (195.2) | 206.5 (199.8) | 199.0 (191.8) | 204.0 (196.8) |
| (NYA) | | **4.7 (4.6)** | **-2.8 (-3.4)** | **2.2 (1.6)** |
| Georg von Neumayer (GVN) | 177.4 (180.3) | 182.4 (185.4) **5.0 (5.1)** | 173.7 (176.4) **-3.7 (-3.9)** | 179.2 (182.0) **1.8 (1.7)** |
| Payerne | 288.8 (291.2) | 293.5 (295.9) | 288.3 (290.9) | 293.0 (295.7) |
| (PAY) | | **4.7 (4.7)** | **-0.5 (-0.3)** | **4.2 (4.5)** |
| Davos | 246.4 (243.9) | 253.4 (250.8) | 244.6 (242.4) | 251.8 (249.6) |
| (DAV) | | **7.0 (6.9)** | **-1.8 (-1.5)** | **5.4 (5.7)** |

