# Peer review of "Revising short and longwave radiation archives in view of possible revisions of the WSG and WISG reference scales: Methods and implications"

_Atmospheric Measurement Techniques, 2017_

## Referee Comment (RC1) · I. Reda (Referee) · 27 Mar 2017

1. Page 1, line 26: Add a reference for the mentioned shortwave range ∼0.3-3 um?

2. Page 1, line 27: Add a reference for the WSG spectral range. I believe that the WSG spectral range is from less than 200 nm to larger than 50 um; this is mentioned in Reda et. al, "Reducing Broadband Shortwave Radiometer Calibration-Bias Caused by Longwave Irradiance in the Reference Direct Beam", Atmospheric and Climate Sciences Vol. 7 (1) January 2017 pp. 36-47. Link: http://file.scirp.org/pdf/ACS_2017011315344478.pdf

3. Page 7, line 35&36: For the BSRN deployed pyrheliometers, change text"virtually all

pyrheliometers are believed to be traceable to the WSG" to the following text "virtually all shortwave pyrheliometers are believed to be traceable to the WSG". Then authors might need to elaborate on why WSG (with a broader spectral range) are used to calibrate shortwave pyrheliometers (with a limited spectral range). Or just open this discrepancy for a discussion within the solar irradiance community/manufacturers.

4. Page 14, Figure 2: the light and dark gray are not distinguishable in the figure, change to two different colors.

---

## Referee Comment (RC2) · Anonymous Referee #2 · 29 Mar 2017

General comments

This manuscript is a great contribution to highlight challenges in and the complexity of high quality radiation measurements. It outlines important issues based on solid data and gives several recommendations on necessary actions derived from their conclusions. However, the question remains how realistic these suggestions or solutions are in the "real" world. Lacking the raw data needed for a recalculation of the radiation values most stations will have to leave their historic data as is. At the present, the added benefit and feasibility of a recalculation of archive data remains questionable. That being said, I nevertheless think it is essential to keep the discussion of these issues open and ongoing. And it is undisputed, that for future studies, the storage of the sensor raw

data is highly desirable.
* * *
Specific comments

- One recommendation I would make is to concentrate on the long-wave fluxes. Introducing also the short-wave flux considerations does not add to the readability of the article. However, if the authors deem it necessary to leave it like that, that's ok with me as well.

- There is a more recent reference for BSRN which should also be cited in chapter 2.3: König-Langlo, G., Sieger, R., Schmithüsen, H., Bücker, A., Richter, F. and Dutton E.G. 2013: The Baseline Surface Radiation Network and its World Radiation Monitoring Centre at the Alfred Wegener Institute. www.wmo.int/pages/prog/gcos/Publications/gcos-174.pdf.

- The amount of abbreviations used is immense, although necessary. To improve intelligibility of the text I suggest to add a list of abbreviations at the end (txt or table form)

- You make it seem, as if the error of 3.5 – 5.4 W/m2 was a general one. However, it only applies to clear-sky night-time measurements, right? You should stress more that these errors are much lower during all-sky 24h measurements (see Table 4), and are in this case within the range of the measurement uncertainty. Recommendation 6 seems therefore currently unnecessary. If raw data will be made available in the future, all studies requesting a IWV correction should be able to apply it to the archived data if necessary.

- Some stations might be especially prone to IWV dependence (dry climate/polar?), other stations in humid climates may not. I am missing a statement on this.

- As you stress the importance of the raw data, and the difficulty of obtaining it I recommend considering the publication of the data files used in this study in a public archive.

Then, these files could also be cited correctly.

- In Figure 3 it becomes clear that the changes in sensitivity (C) between past calibrations/ deployment periods are much larger than the effect of IWV seems to be (compare with Fig. 1). Shouldn't this rather also be a focus of improvement efforts?
* * *
Technical corrections

- Page 1, line 23: According to Google Earth the location of the PMOD in Davos is rather at 46.813°N, 9.844°E, please check. If you give a location it should be as exact as possible in my opinion.

- Page 4, line 33 at the very end: the PMOD CG4-030669 pyrgeometer is only one instrument, right? Then it should read "pyrgeometer" and not "pyrgeometers".

- Page 5, line 7: The sentence starting with "Regardless.." is a little confusing. What do you mean by "here" – also the "later on" is confusing, I'd suggest to rephrase this sentence and just mention the sections where these issues are discussed.

- Page 6, line 11: I do not find a Gröbner and Wacker, 2011 reference in the reference list, so either this reference is missing, or there is a typo in the publication year

- Figures 1 and 2: grey shades are rather hard to differentiate, in Fig 1 I cannot discern three shades of grey, in Fig. 2 the two shades are too similar to each other.

---

## Author Comment (AC1) · 21 Jun 2017

Referee No. 1: Comments and Corrections

Referees comment: 1. Page 1, line 26: Add a reference for the mentioned shortwave range ∼0.3-3 um?

Reply: The wavelength ranges for the shortwave and longwave regions are "broadly defined". Although we have used the word "broad" in the text, we have included a general reference (Petty, 2006) on lines 26 and 31.

[Figure]

Referees comment: 2. Page 1, line 27: Add a reference for the WSG spectral range. I believe that the WSG spectral range is from less than 200 nm to larger than 50 um; this is mentioned in Reda et. al, "Reducing Broadband Shortwave Radiometer Calibration-Bias Caused by Longwave Irradiance in the Reference Direct Beam", Atmospheric and Climate Sciences Vol. 7 (1) January 2017 pp. 36-47. Link: http://file.scirp.org/pdf/ACS_2017011315344478.pdf

Reply: We have added this reference to line 27, and to the reference list, and revised the wavelength range. The sentence now reads:

"The latter is broadly defined as covering the wavelength range $\sim$0.3 − 3 um (Petty, 2006) while the WSG pyrheliometers cover the range from less than 0.2 to above 50 um (Reda et al., 2017)".
* * *
Referees comment: 3. Page 7, line 35&36: For the BSRN deployed pyrheliometers, change text "virtually all pyrheliometers are believed to be traceable to the WSG" to the following text "virtually all shortwave pyrheliometers are believed to be traceable to the WSG".

Reply: This has been changed as recommended.
* * *
Referees comment: Then authors might need to elaborate on why WSG (with a broader spectral range) are used to calibrate shortwave pyrheliometers (with a limited spectral range). Or just open this discrepancy for a discussion within the solar irradiance community/manufacturers.

Reply: This statement was made twice in the original manuscript. The first instance was on Page 2 Line 20 (this is still Page 2 Line 20 in the new manuscript), where we have also added the word "shortwave" as recommended by the Referee. In answer of the comment regarding a mis-match of the spectral ranges, we have added the

following sentence on Page 2 Line 23: "Although the WSG, with its broader spectral range, is used to calibrate shortwave pyrheliometers with their limited spectral range, the spectral mismatch is small in terms of energy. The longwave component is in fact considered to be proportional to the shortwave component for the purpose of the calibration."
* * *
Referees comment: 4. Page 14, Figure 2: the light and dark gray are not distinguishable in the figure, change to two different colors.

Reply: The light and dark shades of grey are now more clearly visible.

---

## Author Comment (AC2) · 21 Jun 2017

Referee No. 2: Comments and Corrections

General comments

This manuscript is a great contribution to highlight challenges in and the complexity of high quality radiation measurements. It outlines important issues based on solid data and gives several recommendations on necessary actions derived from their conclusions. However, the question remains how realistic these suggestions or solutions are in the "real" world. Lacking the raw data needed for a recalculation of the radiation val-

ues most stations will have to leave their historic data as is. At the present, the added benefit and feasibility of a recalculation of archive data remains questionable. That being said, I nevertheless think it is essential to keep the discussion of these issues open and ongoing. And it is undisputed, that for future studies, the storage of the sensor raw data is highly desirable. _____________________ Specific comments
* * *
Referees comment: One recommendation I would make is to concentrate on the long-wave fluxes. Introducing also the short-wave flux considerations does not add to the readability of the article. However, if the authors deem it necessary to leave it like that, that's ok with me as well.

Reply: Our proposal to the Swiss SNF for funds included both the shortwave and longwave ranges. In order to fulfill these obligations, we decided that it would provide a better all-round overview if a discussion of shortwave issues was included in the paper. We hope that this is the case and would therefore prefer to keep the paper in its present form, especially as there is too little material for a separate paper on shortwave issues.
* * *
Referees comment: There is a more recent reference for BSRN which should also be cited in chapter 2.3: König-Langlo, G., Sieger, R., Schmithüsen, H., Bücker, A., Richter, F. and Dutton E.G. 2013: The Baseline Surface Radiation Network and its World Radiation Monitoring Centre at the Alfred Wegener Institute. www.wmo.int/pages/prog/gcos/Publications/gcos-174.pdf.

Reply: We have included the recommended reference at the start of Section 2.3.
* * *
Referees comment: The amount of abbreviations used is immense, although necessary. To improve intelligibility of the text I suggest to add a list of abbreviations at the end (txt or table form)

Reply: We have added a list of abbreviations after the reference list.
* * *
Referees comment: You make it seem, as if the error of 3.5 – 5.4 W/m2 was a general one. However, it only applies to clear-sky night-time measurements, right? You should stress more that these errors are much lower during all-sky 24h measurements (see Table 4), and are in this case within the range of the measurement uncertainty. Recommendation 6 seems therefore currently unnecessary. If raw data will be made available in the future, all studies requesting an IWV correction should be able to apply it to the archived data if necessary.

Reply: These comments are difficult to answer in a single sentence as there are several aspects to be answered, and there may be several small mis-understandings here.

We believe that the Referee is referring to the Abstract on Line 17. We have updated the sentence as we omitted to mention that these refer to all-sky and clear-sky values, respectively, rather than just clear-sky values as mentioned by the Referee. The updated text is in bold below:

"Based on PMOD/WRC calibration archives and BSRN data archives, the downward longwave radiation (DLR) time-series over the 2006 – 2015 periods were analysed at four stations (polar and mid-latitude locations). DLR was found to increase by up to 3.5 and 5.4 W m-2 for all-sky and clear-sky conditions, respectively, after applying a WISG reference scale correction and a minor correction for the dependence of pyrgeometer sensitivity on atmospheric integrated water vapour content".

The above sentence refers to DLR increases at the 4 stations in the study, so we respectfully disagree that this could be interpreted as a "general" increase. We have therefore left the sentence.

Nevertheless, it is true that some DLR increases in Tables 4 and 5 are within the WISG and the IRIS uncertainty ranges of ±2.6 and ±2.0 Wm-2, respectively.
As there is no room in the Abstract to insert more text, we have therefore inserted a sentence concerning the uncertainty on Page 9 Line 3:

"Before considering the overall results in Tables 4 and 5, it should be noted that some of the DLR increases are within the WISG and the IRIS uncertainty ranges of $\pm2.6$ and $\pm2.0$ W m-2, respectively".

Further referees comment: "Recommendation 6 seems therefore currently unnecessary. If raw data will be made available in the future, all studies requesting a IWV correction should be able to apply it to the archived data if necessary".

Based on these constructive comments, we have changed the original text in Recommendation 6 from:

"On the other hand, a revision of longwave BSRN time-series is more difficult as it can only be applied to those pyrgeometers which are traceable to the WISG, and for which raw data are available. A future dedicated study within the framework of BSRN may be more effective in this case".

To:

"On the other hand, a reference scale revision of longwave BSRN time-series is more difficult as it can only be applied to those pyrgeometers which are traceable to the WISG, and for which raw data are available. A nominal correction for the IWV dependence can be applied by the end-user instead, if the measuring pyrgeometer has not had a 3-season calibration at PMOD/WRC. Perhaps of greater importance is whether any increase in average DLR, after all corrections, is within the IRIS uncertainty range of $\pm2.0$ Wm-2. It could be argued that a revision of the historical time-series is therefore not practical, which is true to a certain extent. However, a more general but vital aspect to consider is that the reduction of measurement uncertainty and the increase in accuracy is an important goal of global radiation budget studies".

Referees comment: Some stations might be especially prone to IWV dependence (dry climate/polar?), other stations in humid climates may not. I am missing a statement on this.

Reply: The possible effect on average DLR at other BSRN stations was not adequately discussed, so we have added some text to Page 9 lines 3-9 which discusses this. As average DLR depends on the cloudiness and IWV at each station, we have discussed both aspects rather than just the effect of IWV. We would prefer to avoid a specific statement on IWV dependence and station climatic conditions, and instead just make a general statement at this stage, especially as the IWV correction is relatively small.

The new text is:

"Before considering the overall results in Tables 4 and 5, it should be noted that several of the DLR increases are within the WISG and the IRIS uncertainty ranges of $\pm 2.6$ and $\pm 2.0$ W m-2, respectively. Despite this, if average DLR values are considered to be representative of mid to high-latitude stations, then it implies that similar increases in average DLR may be expected at other BSRN stations, although this will depend on the level of cloudiness and the period of time when IWV < 10 mm. Unfortunately a low-latitude station could not be included in this study but a higher percentage of clear-sky conditions and higher IWV values are likely to occur on average at such locations, resulting in larger increases of average DLR than in Tables 4 and 5. As BSRN stations represent a wide range of dry/humid climates at high/low latitudes, a better estimate of how DLR would change with a reference scale revision and IWV correction will have to await a more detailed study."
* * *
Referees comment: As you stress the importance of the raw data, and the difficulty of obtaining it I recommend considering the publication of the data files used in this study in a public archive. Then, these files could also be cited correctly.

[Figure]

Reply: We agree that this would be the best way forward for the raw data from NYA and GVN used in this study. We will therefore investigate how/whether this can be coordinated with BSRN and station managers from NYA, GVN and PAY.
* * *
Referees comment: In Figure 3 it becomes clear that the changes in sensitivity (C) between past calibrations/deployment periods are much larger than the effect of IWV seems to be (compare with Fig. 1). Shouldn't this rather also be a focus of improvement efforts?

Reply: This is a good point which we have not discussed in enough detail. However, the changes in C are only "apparent". DLR is calculated in Eq. 1 (non-linear) with C but also with time-series of the parameters k1, k2 and k3. Hence, C cannot be compared from deployment period to period, as k1, k2 and k3 are usually different. There is a broad consensus that pyrgeometers give stable DLR measurements, hence we do not believe that an additional focus for improvement is required.

We have therefore changed the sentence on Page 8, line 26 to:

"Figure 3 shows the NYA time-series of C for scenario 4, where different pyrgeometers were deployed for different periods of time (vertical lines). Periods of constant C within each pyrgeometer deployment period occur when IWV > 10 mm, while periods of variable C occur when IWV < 10 mm. Note that any differences in C from deployment period to period for the same pyrgeometer, reflect the non-linear nature of Eq. 1 (ki time-series are also required to calculate DLR) rather than any instability in pyrgeometer measurements".

Table 1 also has a revised footnote to take this into consideration.
* * *
Technical corrections

Referees comment: Page 1, line 23: According to Google Earth the location of the PMOD in Davos is rather at 46.813N, 9.844E, please check. If you give a location it should be as exact as possible in my opinion.

Reply: The coordinates we use, stem from Swiss Metas determinations in the past. We prefer using these values rather than those from Google for a number of reasons. However, instead of giving our location to 2 decimal places, we have now given the coordinates to 4 decimal places. This is now: 46.8143°N, 9.8458°E.
* * *
Referees comment: Page 4, line 33 at the very end: the PMOD CG4-030669 pyrgeometer is only one instrument, right? Then it should read "pyrgeometer" and not "pyrgeometers".

Reply: This has been corrected as recommended.
* * *
Referees comment: Page 5, line 7: The sentence starting with "Regardless.." is a little confusing. What do you mean by "here" – also the "later on" is confusing, I'd suggest to rephrase this sentence and just mention the sections where these issues are discussed.

Reply: We agree with the Referee. In fact, the sentence is not really required, and we have therefore removed it to avoid any confusion.
* * *
Referees comment: Page 6, line 11: I do not find a Gröbner and Wacker, 2011 reference in the reference list, so either this reference is missing, or there is a typo in the publication year.

Reply: Sorry, this reference was missing but has now been changed to Wacker et al. (2011), and included in the reference list.

Referees comment: Figures 1 and 2: grey shades are rather hard to differentiate, in Fig 1 I cannot discern three shades of grey, in Fig. 2 the two shades are too similar to each other.

Reply: The light and dark shades of grey are now more clearly visible.

[Figure]

[Figure]

**Fig. 1.**